# The p38/MK2 Pathway Functions as Chk1-Backup Downstream of ATM/ATR in G_2_-Checkpoint Activation in Cells Exposed to Ionizing Radiation

**DOI:** 10.3390/cells12101387

**Published:** 2023-05-14

**Authors:** Daxian Luo, Emil Mladenov, Aashish Soni, Martin Stuschke, George Iliakis

**Affiliations:** 1Institute of Medical Radiation Biology, University Hospital Essen, University of Duisburg-Essen, 45147 Essen, Germany; luodaxian@hotmail.com (D.L.); emil.mladenov@uk-essen.de (E.M.); aashish.soni@uk-essen.de (A.S.); 2Division of Experimental Radiation Biology, Department of Radiation Therapy, University Hospital Essen, University of Duisburg-Essen, 45147 Essen, Germany; martin.stuschke@uk-essen.de; 3German Cancer Consortium (DKTK), Partner Site University Hospital Essen, 45147 Essen, Germany; 4German Cancer Research Center (DKFZ), 69120 Heidelberg, Germany

**Keywords:** checkpoints, G_2_-checkpoint, MK2, p38, Chk1, MAP-kinases, ionizing radiation

## Abstract

We have recently reported that in G_2_-phase cells (but not S-phase cells) sustaining low loads of DNA double-strand break (DSBs), ATM and ATR regulate the G_2_-checkpoint epistatically, with ATR at the output-node, interfacing with the cell cycle through Chk1. However, although inhibition of ATR nearly completely abrogated the checkpoint, inhibition of Chk1 using UCN-01 generated only partial responses. This suggested that additional kinases downstream of ATR were involved in the transmission of the signal to the cell cycle engine. Additionally, the broad spectrum of kinases inhibited by UCN-01 pointed to uncertainties in the interpretation that warranted further investigations. Here, we show that more specific Chk1 inhibitors exert an even weaker effect on G_2_-checkpoint, as compared to ATR inhibitors and UCN-01, and identify the MAPK p38α and its downstream target MK2 as checkpoint effectors operating as backup to Chk1. These observations further expand the spectrum of p38/MK2 signaling to G_2_-checkpoint activation, extend similar studies in cells exposed to other DNA damaging agents and consolidate a role of p38/MK2 as a backup kinase module, adding to similar backup functions exerted in p53 deficient cells. The results extend the spectrum of actionable strategies and targets in current efforts to enhance the radiosensitivity in tumor cells.

## 1. Introduction

In higher eukaryotes, the risks posed to the genome by DNA double-strand breaks (DSBs) are mitigated by a network of signaling pathways collectively termed the DNA damage response (DDR). The DDR detects DSBs and coordinates a wide spectrum of cellular responses, including checkpoint activation and DSB repair [1,2,3]. Three major cell cycle checkpoints are activated by DSBs: The G_1_-checkpoint is activated in cells irradiated in G_1_-phase, or cells dividing with unrepaired DNA damage, and delays cell progression into S-phase and thus the inception of DNA replication. The intra-S-phase checkpoint is activated in cells irradiated during DNA replication, or cells progressing from G_1_-phase with unrepaired DNA damage and reduces DNA replication activity by suppressing the activation of new replication origins. The G_2_-checkpoint is activated in cells irradiated in G_2_-phase or cells entering G_2_-phase from S-phase with unrepaired DNA damage and delays progression into M-phase [4,5,6]. Checkpoints ensure that repair takes place before inception of major cell division events, e.g., DNA replication and mitosis that are associated with changes in chromatin organization interfering with DSB repair. Therefore, delays in cell cycle progression induced by checkpoint activation maximize genomic stability in cells sustaining DNA damage.

Checkpoints inhibit cell cycle progression by suppressing the activation of the cyclin-dependent kinases (CDKs) that catalyze cell cycle transitions or by activating their inhibitors (CDKis) [1,2,3]. Two members of the PI3 kinase family of protein kinases are located at the apex of the signaling network initiated by DSBs and are directly implicated in checkpoint activation, ATM (ataxia telangiectasia mutated), and ATR (ATM and Rad3-related). They interact with the cell cycle engine by activating the checkpoint kinases Chk1 and Chk2, which inhibit cell cycle progression by inactivating members of the Cdc25 family of phosphatases—the key activators of CDKs. The Cdc25 family consists of three isoforms: Cdc25A, Cdc25B, and Cdc25C. Chk1/Chk2 mediated phosphorylation of Cdc25A causes its ubiquitin-mediated degradation, while equivalent phosphorylation of Cdc25B and Cdc25C causes sequestration from the nucleus to the cytoplasm through binding to 14-3-3 proteins preventing thus CDK activation in the nucleus and cell cycle progression [7,8,9,10,11,12].

We have recently reported that at ionizing radiation (IR) doses, between 1 and 2 Gy, that induce approximately 40–80 DSBs in the genome of G_2_-phase irradiated cells [13], ATM and ATR regulate the G_2_-checkpoint epistatically, with ATR at the output-node, interfacing with the cell cycle machinery predominantly through Chk1 [14]. Strikingly, at low IR doses, ATM and ATR epistatically also regulate resection and inhibition of either activity fully suppresses resection. At higher IR doses, the tight ATM/ATR coupling relaxes and independent outputs to G_2_-checkpoint and resection manifest [14]. Consequently, both kinases must be inhibited to fully suppress checkpoint activation and resection. DNA-PKcs integrates into the ATM/ATR module by regulating resection, with defects in DNA-PKcs causing hyper-resection and G_2_-checkpoint hyperactivation [15]. Hyper-resection and G_2_-checkpoint hyperactivation are absent from other c-NHEJ mutants, ruling out a general c-NHEJ suppression as the main culprit for this effect and suggesting regulatory functions for DNA-PKcs in the overall response [15]. Therefore, we proposed that a subset of DSBs that increases with a decreasing IR dose [16] is shunted by DNA-PKcs from c-NHEJ to resection-dependent DSB-processing that occurs under the regulatory supervision of the ATM/ATR module [13,14,15].

In the above study [14], we noticed that the DNA-PKcs/ATM/ATR module suppresses cell cycle progression from G_2_- to M-phase solely through Chk1 activation. Chk2, while activated to normal levels, fails to contribute to the suppression of cell cycle progression. However, although inhibition of ATR nearly completely abrogates the G_2_-checkpoint, inhibition of Chk1 with UCN-01 generates a strong but only partial suppression of the checkpoint. This result suggested to us that additional kinases are involved in the transmission of the checkpoint signal from DNA-PKcs/ATM/ATR to the cell cycle engine. Along these lines, we found it intriguing that a member of the MAP kinase family, p38 MAPK (henceforth p38), and its direct downstream target MAPKAP kinase-2 (henceforth, MK2) have been implicated in the regulation of various aspects of a G_2_-checkpoint activated either in cells exposed to ultraviolet (UV) radiation or to selected chemotherapeutic agents [17,18,19,20,21,22,23,24,25,26]. Furthermore, MK2 is inhibited by UCN-01 with comparable efficacy as Chk1 [18], raising the possibility of undetected contributions to our UCN-01 results and increasing the likelihood of MK2 contributions to our observations [13,14,15].

The MAPK superfamily includes the extracellular signal-regulated kinase (ERK), the c-Jun N-terminal kinase (JNK), and the p38 MAPK. While the ERK pathway is activated by mitogenic stimuli, p38 and JNK are only weakly activated by growth factors and respond much stronger to cellular stress; JNK and p38 kinases are, therefore, also referred to as stress-activated protein kinases (SAPK). Each MAPK signaling pathway mainly consists of three components, a MAPK kinase kinase kinase (MAP3K), a MAPK kinase (MAP2K), and finally, the MAPK that signals to different effectors depending on the context [27,28,29,30,31,32]. Furthermore, p38 kinases are proline-directed serine/threonine kinases that are themselves canonically activated by dual Thr180 and Tyr182 phosphorylation in the activation loop and are found in all eukaryotes from yeast to humans [27,28,29,30,31].

There are four homologs of p38: p38α, p38β, p38γ, and p38δ. In spite of their structural similarities, p38 kinases differ with respect to their downstream targets and sensitivity to chemical inhibitors [27], while p38α and p38β are ubiquitously expressed, the expression of p38γ and p38δ is more tissue-specific [27]. The p38 family, particularly p38α, is known to be activated by DNA damage and to support pro-survival responses [29,33,34,35,36,37], including checkpoint activation together with its downstream target MK2 [17,20,27,28,29,30,31,38]. Notably, however, with one exception [39], p38 proteins could not be implicated in G_2_-checkpoint control in cells exposed to IR [20,38].

Here, we show that Chk1 inhibitors more specific than UCN-01 exert an even weaker effect on the G_2_-checkpoint than previously observed, while p38α/MK2 inhibitors are on their own completely ineffective, as already reported [18,20]. Strikingly, however, the strong residual G_2_-checkpoint following Chk1 inhibition is fully dependent on the p38α/MK2 pathway. We propose that in cells exposed to IR, the p38α/MK2 pathway is wired to function on the G_2_-checkpoint as a backup to Chk1 signaling.

## 2. Materials and Methods

### 2.1. Cell Culture

Human lung epithelial carcinoma A549 cells were grown in McCoy’s 5A medium supplemented with newborn calf serum and 0.5 μg/mL Fe^2+^ (FeSO_4_·7H_2_O) and antibiotics. Parental colon cancer cell line HCT116 (HCT116-p53^+/+^) and their p53-deficient counterpart (HCT116-p53^−/−^) were grown in McCoy’s 5A medium supplemented with 10% fetal bovine serum (FBS) and antibiotics. Immortalized human fibroblasts 82-6-hTert cells were grown in MEM medium supplemented with 10% FBS, 1% non-essential amino acids (NEAA), and antibiotics. Exponentially growing cells were passaged every second day while maintaining a maximum confluence of no more than ~80%. All cell lines were grown in flasks at 37 °C in an atmosphere of 95% air and 5% CO_2_.

### 2.2. Inhibitors

Kinase inhibitor stocks were prepared by dissolving the chemicals in dimethyl sulfoxide (DMSO). Inhibitors were administrated to the cells 1 h before irradiation. Control groups were treated only with DMSO. The inhibitors were kept until the cells were collected for analysis. An ATM inhibitor, KU55933 (ATMi, Haoyuan ChemExpress, Shanghai, China), was used at 10 μM final concentration. The Chk2 inhibitor (Chk2 Inhibitor-II/BML-277, Chk2i, Calbiochem, Darmstadt, Germany) was used at 400 nM final concentration. The ATR inhibitor, Berzosertib (VE-822) (ATRi, Selleckchem, Planegg, Germany), was used at 300 nM final concentration. The Chk1 inhibitor, PF477736 (Chk1i, Selleckchem, Planegg, Germany), was used at 300 nM final concentration. The MK2 inhibitor, PF3644022 (MK2i, TOCRIS, Wiesbaden-Nordenstadt, Germany), was used at 1 µM final concentration. The p38α inhibitor, Ralimetinib, LY2228820 (p38αi, Selleckchem, Planegg, Germany), was used at 1 µM final concentration. The working inhibitor concentrations were experimentally adjusted for maximum effect by semi-quantitative Western blot analysis using phosphorylation levels of key targets as a proxy (see corresponding Figures). We selected concentrations causing maximum kinase inhibition without detectably changing cell cycle distribution. The inhibitor concentrations determined in this way are well within the range of concentrations reported in the literature for several cell lines: ATRi, VE-822 (80–400 nM), Chk1i, PF477736 (45–300 nM), p38αi, LY2228820 (0.2–1 μM), and MK2i, PF3644022 (0.16–10 μM).

### 2.3. Irradiation

Irradiation of cells was performed with an X-ray machine (GE Healthcare, Solingen, Germany), operating at 320 kV with a 1.65 mm Al filter at a distance of 500 mm from the source and a dose rate of 3.5 Gy/min. An even exposure to radiation was ensured by rotating the radiation table. Proper dosimetry of the X-ray machine is performed regularly by Fricke dosimetry. To avoid temperature fluctuations, cells were irradiated in a thin-walled aluminum box filled with pre-warmed water. After irradiation, cells were transferred to 37 °C and were collected according to the requirements of the experimental protocol (see below).

### 2.4. Two Parametric Flow Cytometry Analysis of Mitotic Index Using Propidium Iodide (PI) and H3-pS10 Staining

Two parametric flow cytometry analysis was employed to simultaneously measure DNA content by PI and cells in mitosis by detection of phosphorylated Histone H3 at Serine 10 (H3-pS10). Briefly, cells were fixed overnight in 70% ethanol at 4 °C and were permeabilized for 15 min in ice-cold PBS supplemented with 0.25% Triton X-100. Cell pellets were incubated in PBG blocking buffer (0.2% gelatin and 0.5% bovine serum albumin in PBS) for 1 h at RT, followed by incubation with anti-H3-pS10 antibody (rabbit polyclonal, Abcam PLC, Oxford, UK) for 1 h at RT (See Appendix A for the ranges of the utilized dilutions). Cells were washed with PBS and incubated in AlexaFluor 488-conjugated goat anti-rabbit-IgG (Thermo Fisher Scientific, Waltham, MA, USA) diluted in PBG for 1 h at RT in the dark. Finally, DNA was stained with PI at 37 °C for 15 min. All incubation steps were performed under gentle agitation. Analysis was carried out using a Gallios flow cytometer (Beckman Coulter, Krefeld, Germany). Proper gating was applied to select H3-pS10 positive events that represented cells in mitosis (Appendix A). The mitotic index (MI) was calculated as the fraction of cells in mitosis and is shown normalized to the MI of non-irradiated control groups. The ranges of MIs (RAW MIs) for non-irradiated control cells used for normalization are given in the legends of the corresponding figures.

### 2.5. Three Parametric Flow Cytometry Analysis of Mitotic Index Using PI, EdU, and H3-pS10 Staining

Three parametric flow cytometry analysis was developed to measure the activation and propagation of the G_2_-checkpoint in cells irradiated in S-phase of the cell cycle, and in the G_2_ phase. The method is based on the parallel quantification of PI, indicating cellular DNA content, ethylene-deoxyuridine (EdU) marking S-phase cells at the time of irradiation, and H3-pS10 marking mitotic cells. Briefly, exponentially growing cells were pulse-labeled with 2 μM EdU for 30 min prior to inhibitor treatment and/or irradiation. After the incubation period, EdU was washed away by pre-warmed PBS. Pre-warmed, pH-adjusted EdU-free growth medium, was then added to cells. At the indicated time intervals (0.5, 1, 2, 4, 6, and 8 h) after irradiation with 2 Gy (if not indicated otherwise), cells were collected by trypsinization and were fixed for 15 min at RT in 3% paraformaldehyde (PFA), 2% sucrose, dissolved in PBS. Cells were incubated in ice-cold PBS containing 0.25% Triton™ X-100 for 15 min and blocked overnight with PBG blocking buffer at 4 °C. Subsequently, cells were incubated in primary anti-H3-pS10 antibody (rabbit, polyclonal, and Abcam) for 1 h at RT (See Appendix A for the ranges of the utilized dilutions). Cells were washed with PBS and incubated in AlexaFluor 488-conjugated goat anti-rabbit-IgG (Thermo Fisher Scientific, Waltham, MA, USA) for 1 h at RT. Subsequently, cells were washed with PBS, and the EdU signal was developed using an EdU staining kit (Thermo Fisher Scientific, Waltham, MA, USA) according to the manufacturer’s instructions. Finally, to stain DNA with PI, cells were incubated with PI staining solution (4 mg/mL PI, 10 mg/mL RNAse, dissolved in PBS) at 37 °C for 30 min. The RNAse treatment eliminates the interference of RNA with the DNA content acquisition by FACS. Flow cytometry analysis was carried out in a Gallios flow cytometer (Beckman Coulter, Krefeld, Germany). All incubation steps were performed under gentle agitation. Proper gating for EdU positive (EdU^+^, G_2_-cells) and EdU negative (EdU^−^, G_2_-cells) cells was applied to analyze the cell population of interest (Appendix A). Kaluza 2.1 software was used for data analysis.

### 2.6. SDS-PAGE and Western Blot Analysis

SDS-PAGE and Western blot analysis were carried out according to previously published protocols [13,14,16]. Briefly, cells were collected by trypsinization, washed in ice-cold PBS, and lysed in ice-cold RIPA buffer (Thermo Fisher Scientific, Waltham, MA, USA) containing protease and phosphatase inhibitor cocktails (Thermo Fisher Scientific, Waltham, MA, USA), as recommended by the manufacturer. Protein lysates were resolved in SDS-PAGE gels and were transferred to nitrocellulose membrane for Western blot analysis. The primary antibodies were anti-Chk1 (G-4) (Santa Cruz Biotechnology, Heidelberg, Germany), anti-pChk1-S345, anti-pChk1-S296, anti-HSP27, anti-pHSP27-S82, anti-MK2 and anti-pMK2-T334 (all from Cell Signaling Technology, Leiden, The Netherlands), anti-Ku80 (GeneTex, Irvine, CA, USA), anti-Ku70 (N3H10) (GeneTex, Irvine, USA), and anti-GAPDH (MERCK, Darmstadt, Germany) at the corresponding dilutions (See Appendix A for details). The secondary antibodies were anti-mouse and anti-rabbit IgG conjugated with IRDye680 and IRDye800 (LI-COR Biosciences, Bad Homburg, Germany) at 1:10,000 dilution. Immunoblots were scanned on the Odyssey infrared scanner (LI-COR Biosciences, Bad Homburg, Germany). The RAW Western blot results are shown in Appendix A.

## 3. Results

### 3.1. Chk1 Inhibition Only Partially Suppresses G_2_-Checkpoint at Low IR Doses

We recently reported that G_2_–phase cells exposed to doses below 2 Gy of X-rays activate a G_2_-checkpoint that is equally dependent on ATM and ATR in the sense that inhibition of either kinase almost fully abrogates its activation [13,14]. This observation was unexpected as it is widely thought that ATM is the main regulator of the G_2_-checkpoint [4] and that ATR has a much smaller contribution [40,41,42,43,44] that may increase only under special circumstances [45,46]. These observations led us to propose that ATM and ATR regulate G_2_-checkpoint as an integrated module in which the two kinases are functionally, and likely also physically paired and operating in an epistatic manner [13,14].

ATM and ATR initiate the checkpoint by activating the checkpoint kinases Chk2 and Chk1, respectively, which in turn phosphorylate and inhibit the activities of Cdc25s [4,7,8,9,10,11,12,13,14]. Notably, when we analyzed the contributions of Chk2 using a highly specific Chk2 inhibitor (Chk2 inhibitor II; BML-277), we were unable to detect suppression of checkpoint activation [44,47,48,49,50]. On the other hand, UCN-01, a relatively non-specific inhibitor of Chk1 [18], strongly regulated by ATR, caused a marked, albeit incomplete, suppression of the G_2_-checkpoint [14]. These observations suggested that the ATM-mediated activation of Chk2 is not wired to directly mediate cell cycle arrest under these conditions and that this task is transferred by ATM to ATR.

It appeared surprising that the effects of ATR and Chk1 inhibitors were not equivalent, which suggested unknown inputs in the regulation of the G_2_-checkpoint that warranted further studies. This need was further reinforced by the relatively low specificity of UCN-01 for Chk1 [18], and by the continuous development for clinical testing of new and more specific Chk1 and ATR inhibitors that allow further testing of the generality of our earlier observations. An in-depth analysis of these aspects of the G_2_-checkpoint regulation is the focus of the experiments outlined below.

The two-parameter flow cytometry method combining DNA and histone H3 phospho-Serine 10 (H3-pS10) detection to identify mitotic cells [40,44,51] and to quantitate the mitotic index (MI) is outlined in Appendix A. It allows us to specifically analyze the response of cells irradiated in the G_2_-phase of the cell cycle, which, as we showed, is essential for analysis of the mechanistic underpinning of the checkpoint response [14]. Indeed, we reported that cells irradiated in S-phase utilize a very different form of ATM and ATR crosstalk to regulate the G_2_-phase checkpoint [13].

As a first step in our investigation, we studied checkpoint activation in proliferating hTert immortalized normal human fibroblasts 82-6 (82-6 hTert) to a low IR dose (2 Gy) and measured the effects of ATR inhibition using new compounds. Figure 1A shows a precipitous drop of the normalized MI in these cells at 1 h that reflects the prompt activation of the G_2_-checkpoint. As mainly cells in G_2_-phase reach M-phase within the observation time interval, the effect reflects exclusively the response of cells that were irradiated in G_2_-phase. The checkpoint remains active for up to 4 h, but cells start re-entering mitosis at later times, indicating recovery from the checkpoint that is nearly complete at 8 h.

When we treat cells with the ATR-specific inhibitor Berzosertib (VE-822) (IC50 of 19 nM), cells progress into mitosis almost uninterrupted, indicating a nearly complete abrogation of the checkpoint that is practically identical to that previously reported for VE-821 [14], an ATR inhibitor from the same family with an IC50 of 13 nM (dotted line in Figure 1A) [52,53,54]. Notably, all stages of the G_2_-checkpoint, including initiation, maintenance, and recovery, are abrogated and irradiated cells enter mitosis practically without delay. Similar observations are made with the human lung carcinoma cell line, A549 (Appendix A). The strong inhibition by VE-822 of the ATR-dependent phosphorylation of Chk1 on S345 in 82-6 hTert cells confirms the effectiveness of the inhibitor under our conditions (Appendix A). Collectively, these results consolidate the complete control by ATR of the G_2_-checkpoint in G_2_-phase cells exposed to 2 Gy of IR.

Next, we investigated the role of Chk1 in the transmission of the ATR signal. Since UCN-01 has an IC50 of 7 nM for Chk1 but also targets, with comparable affinity, several other cellular kinases of the DNA damage response [18], we searched for more specific inhibitors. PF-477736 has an IC50 of 0.49 nM for Chk1 and a ~100-fold higher IC50 of 47 nM for Chk2; moreover, compared to UCN-01, it inhibits fewer cellular kinases and with higher IC50 than Chk1 and is, therefore, a much more specific, ATP-competitive, Chk1 inhibitor [55]. Appendix A show that PF477736 at 300 nM strongly inhibits Chk1 activity in 82-6 hTert and A549 cells, as measured by the inhibition of its autophosphorylation at S296. Strikingly, Figure 1A shows that when tested in 82-6 hTert cells, PF477736 at a concentration of 300 nM only slightly suppresses checkpoint activation, although it markedly accelerates its recovery. This effect is significantly smaller than that of ATR inhibition and, most significantly, lower than that of UCN-01 [14] (broken line in Figure 1A). Similar results are also obtained with A549 cells, which show more prominent differences only at the later time points after irradiation (2, 4, and 8 h) (Appendix A). Moreover, the inhibitors show no dramatic effect on the progression of non-irradiated cells through mitosis, as their MI remained stable in the course of 8 h investigation (Appendix A)

This result is significant because it confirms our postulate that upon ATR activation, downstream kinases, in addition to Chk1, must be able to contribute to checkpoint activation. It also suggests that the low specificity of UCN-01 partly masks the contributions of these kinases (possibly by inhibiting them) and overestimates the effect of “Chk1” inhibition. These “unknown” kinases may either share work with Chk1 and operate in parallel, or they may kick in as backup once Chk1 is inhibited. In the following sections, we outline experiments testing the contributions of the p38α/MK2 pathway in this response and establishing its functional coordination with Chk1.

### 3.2. Chk1 Is Neither Assisted Nor Backed-Up by Chk2 in the Regulation of the G_2_-Checkpoint

Chk1 and Chk2 are structurally unrelated but have overlapping serine/threonine kinase functions, activated in response to various genotoxic stresses. As already extensively discussed [13,14], crosstalk between ATR-Chk1 and ATM-Chk2 pathways exits. Therefore, we considered it possible that Chk2 partially compensates for Chk1 inhibition under certain conditions. As previous work along these lines was carried out with UCN-01, we repeated the experiments with the more specific Chk1 inhibitors introduced here. Unlike PF477736, 400 nM Chk2 inhibitor II (BML-277) has, as already reported [14], a small effect only on the recovery of the G_2_-checkpoint activated after 2 Gy of IR (Figure 1B). Combined inhibition of Chk1 and Chk2 fails to further suppress the checkpoint in a statistically significant manner, suggesting that the ATM-Chk2 signal axis fails to directly connect to the cell cycle machinery and that ATR (and ATM) suppresses cell cycle progression using other downstream effectors when Chk1 is inhibited. Similar conclusions can be drawn from the results obtained with A549 cells (Appendix A).

### 3.3. The p38/MK2 Pathway Regulates the G_2_-Checkpoint, but Only When Chk1 Is Inhibited

As we discussed in the Introduction, activation of the G_2_-checkpoint after exposure of cells to UV and various chemotherapeutic agents requires p38/MK2 signaling. However, effects on the G_2_-checkpoint activated after IR have not been found [17,20] or are restricted to p38γ [39]. To investigate whether and under what circumstances the p38α/MK2 pathway is involved in the G_2_-checkpoint after exposure to IR, a novel, potent, and selective ATP-competitive small molecule inhibitor of p38α, LY2228820 (IC50 of 7 nM), was tested in experiments similar to those described above. In addition, a potent and selective ATP-competitive small molecule inhibitor of MK2 [19,26], PF3644022, was tested (IC50 of 5.2 nM) in parallel.

To validate the efficacies of LY2228820 and PF3644022, we measured, using Western blotting, impact of the former on p38α activity by measuring phosphorylation of MK2 on Threonine 334 (T334) and of the latter on MK2 by analyzing HSP27 phosphorylation at S82, a known target of the MK2 kinase [56,57,58]. Figure 2A,B show that IR activates the p38α and MK2 kinases, and that LY2228820 and PF3644022, respectively, strongly suppress this activation in both 82-6 hTert and A549 cells. On the basis of these results, we selected 1 µM as the working concentration for both inhibitors.

Notably, the results on checkpoint activation show that inhibition of p38α, either in 82-6 hTert or A549 cells with LY2228820 (p38αi) (Figure 2C,D) generates no detectable inhibitory effect, as already reported [17,20]. The MK2 inhibitor PF3644022 actually renders the G_2_-checkpoint slightly stronger (Figure 2C,D).

The contributions of the p38α/MK2 pathway to the G_2_-checkpoint in cells exposed to chemotherapeutic agents is stronger in p53 deficient cells [17,18,19]. We compared therefore HCT116 p53 deficient cells with their wild-type counterparts in experiments similar to those of Figure 2. The results in Appendix A show that inhibition of p38α or MK2 kinases fails to suppress G_2_-checkpoint activation also in HCT116 cells and that the p53 defect fails to alter the outcome.

We next inquired whether p38α/MK2 signaling contributes to the G_2_-checkpoint when Chk1 is inhibited—as a quasi-backup. Strikingly, combined treatment of 82-6 hTert cells exposed to 2 Gy of IR with Chk1 and p38α inhibitors causes a strong abrogation of the checkpoint (Figure 3A). Suppression is also observed in A549 cells, but the effect is less pronounced than in 82-6 hTert cells (Figure 3B). The MK2 inhibitor also markedly suppresses the G_2_-checkpoint when given together with the Chk1 inhibitor in 82-6 hTert cells (Figure 3C), and here again a smaller effect is observed in A549 cells (Figure 3D). This is evidence that when Chk1 is suppressed, the p38α/MK2 pathway is recruited to activate and maintain the G_2_-checkpoint after exposure of cells to low doses of IR.

We examined possible crosstalk between Chk2 and p38α/MK2 signaling. However, combination of Chk2 with p38α/MK2 inhibitors leaves the checkpoint intact in 82-6 hTert cells (Figure 4A,B). Even combined Chk1, Chk2 and p38α/MK2 inhibition in 82-6 hTert cells fails to generate excess effect, as compared to that generated by inhibition of only Chk1 plus p38α/MK2 (Figure 4C,D). Similar results are obtained with A549 cells (Appendix A).

To further validate the effects of combined inhibition of Chk1 and p38α/MK2 on the G_2_-checkpoint and to also examine possible differences in this regulation between cells irradiated in G_2_-phase and cells irradiated in S-phase, we pulse-labelled 82-6 hTert cells with EdU for 30 min, before IR exposure, and employed three parametric flow cytometry to separately quantitate the regulation of the G_2_-checkpoint in cells irradiated during the S-phase and which therefore are EdU positive (EdU^+^) and cells irradiated in G_2_-phase, which therefore are EdU negative (EdU^−^) (Appendix A). EdU^+^ cells in G_2_-phase during analysis represent those irradiated in S-phase that have post-irradiation progressed to G_2_-phase. Appendix A shows the gates adopted to measure H3pS10 signal from G_2_-phase (EdU^−^) irradiated cells, and the fraction of G_2_-phase cells for cells irradiated in S-phase (EdU^+^).

Appendix A shows that when EdU^−^ cells are followed as a function of time after EdU labelling (and eventual irradiation), cells divide and the G_2_-compartment is naturally depleted of cells, in the absence or IR, within 8 h. Irradiation transiently stops division and delays cell depletion from the G_2_-phase compartment. As expected from the results in Figure 1B, inhibition of Chk1 has no effect at early times but accelerates after 4 h cell depletion from the G_2_-phase compartment (Appendix A). Additionally, the results obtained after combining Chk1 with p38α/MK2 inhibition (Appendix A) confirm the results in Figure 3.

Follow-up of MI as a function of time after IR for EdU^−^ cells clearly shows the depletion of non-irradiated mitotic cells and the activation and recovery of the G_2_-checkpoint in irradiated cells (Figure 5A). In irradiated EdU^−^ cells the MI would also drop to zero, of course, but requires an additional of 4–8 h after checkpoint recovery. The results in Figure 5B–D confirm the abrogation of the checkpoint following inhibition of Chk1 and/or p38α/MK2 kinases. These results are similar to those obtained in cells analyzed in the absence of EdU but are significant because they exclude potentially spurious contributions or effects arising from S-phase irradiated cells [13,14].

G_2_-checkpoint responses of S-phase irradiated cells can be analyzed by following the increase in EdU^+^ cells specifically in the G_2_-phase compartment (Figure 6) [13,14]. The fraction of EdU^+^, G_2_-phase cells at 0 h (~10%) in non-irradiated cultures reflects cells at the S/G_2_ border during the 30 min of labeling with EdU. The modest gradual increase after 2 h is caused by the increased occupancy with cells of earlier segments of the cell cycle in an exponentially growing population (naturally resulting from the fact that one dividing mitotic cell generates two G_1_-cells), and possibly also by perturbations in DNA replication owing to experimental manipulations that generate a form of para-synchrony in the cell culture [59]. After 2 Gy, the ensuing arrest of S-phase irradiated cells in G_2_-phase increases further the fraction of cells in G_2_-phase and is therefore a reflection G_2_-checkpoint activation (generating an ~3 h delay when arbitrary measured at the 20% fraction) (Figure 6A).

Exposure to Chk1 inhibitor decreases the accumulation of irradiated cells in G_2_-phase in this form of analysis, suggesting a suppression of the checkpoint even for cells irradiated in S-phase (Figure 6B). However, when considering the effect of the Chk1 inhibitor on non-irradiated cells, a clear residual checkpoint is noted (Figure 6B) which is also in line with the observations with cells irradiated in G_2_-phase (Figure 1). Notably, the practically identical accumulation in G_2_-phase of both irradiated and non-irradiated cells exposed to combinations of Chk1/p38α or Chk1/MK2 inhibitors (Figure 6C,D) is compatible with the interpretation that these combined treatments suppress the G_2_-checkpoint response in S-phase irradiated cells, as they also do in G_2_-phase irradiated cells (Figure 2). We conclude that the mechanistic underpinnings of the cooperation between Chk1 and p38α/MK2 in the regulation of the G_2_-checkpoint are similar for cells irradiated in G_2_- and S-phase of the cell cycle. This contrasts the ATM/ATR crosstalk that is radically different in these two cell populations [13,14].

### 3.4. Upstream Kinases Involved in the Activation of the p38a/MK2 Pathway

We next inquired whether p38α and MK2 kinase activation, as measured by Western blotting, is ATM- and/or ATR-dependent. When 82-6 hTert cells are exposed to 10 Gy of IR and analyzed 1 h later, strong activation of p38 is observed demonstrated by the appearance of a strong pMK2-T334 band (Figure 7A). The intensity of this band is notably decreased after incubation with ATMi and strongly suppressed after incubation with ATRi. Combined ATMi and ATRi inhibition almost completely abrogates pMK2-T334 formation and incubation with caffeine, a non-specific pan-PIKK inhibitor, generates similar effects. We conclude that activation of p38α is primarily ATR-dependent with detectable inputs from ATM. It remains possible however that the observed ATM-dependence is stronger at the high dose of 10 Gy used here and may diminish at the lower doses of 2 Gy used in the G_2_-checkpoint experiments. It is challenging to reliably measure p38α and MK2 activation at such low doses.

p38α activation decreases after incubation with Chk1i (Figure 7A), as shown by the intensity decrease in the pMK2-T334 signal but is also almost completely suppressed by ATRi. Notably, under Chk1 inhibition conditions, ATMi relieves the Chk1i-dependent suppression of pMK2-T334 signal. Note that Chk1 activity is strongly suppressed after exposure to Chk1i, as indicated by the strong reduction in pChk1-S296 signal. We conclude that p38α activation remains ATR dependent even after inhibition of Chk1, the condition that unleashes its backup role in G_2_-checkpoint control (see previous sections). However, the results in Figure 7A also reveal interdependencies among the different kinases tested in the activation of p38α that will require further investigations.

Downstream of p38α, activation of MK2 is also directly evident after exposure to 10 Gy by the phosphorylation of HSP27 at S82. However, here inhibition of ATM, ATR and even their combination, or caffeine, have a rather small effect. Notably, after incubation with Chk1i, MK2 appears less suppressed than p38α, and now ATMi and ATRi generate robust inhibitory effects. These results suggest that MK2 activation after exposure to IR remains ATM and ATR dependent and show that this dependency becomes consolidated after inhibition of Chk1—in line with the backup function on G_2_-checkpoint activation shown above. Western blot analysis confirmed that treatment of 82-6 hTert cells with p38αi or MK2i, results in a decrease in pMK2-T334 and pHSP27-S82 phosphorylation, indicating inhibition of p38α and MK2 kinase activities (Appendix A). 

Analogous experiments with A549 cells (Appendix A) show qualitatively similar dependencies of p38α and MK2 activation on ATM and ATR, but the overall effects after their inhibition are less pronounced—again in line with the smaller effect on G_2_-checkpoint of p38α and MK2 inhibition observed in this cell line following treatment with Chk1i. 

Finally, G_2_-checkpoint experiments revealed that combined treatment of 82-6 hTert fibroblasts with ATR and p38α inhibitors generate an effect stronger than ATR inhibition alone (Figure 7B). This suggests that in irradiated cells p38α can receive activation signals from kinases other than ATR. On the other hand, combined inhibition of ATR with the MK2 inhibitor fails to potentiate the ATR effect, which suggests that p38α may have alternative ways to connect to the checkpoint when MK2 is inhibited (Figure 7C). ATM and p38α inhibition generate a response similar to that of ATM inhibition alone, suggesting epistatic or independent functions (Figure 7D) and the same holds true for combined treatment with ATM and MK2 inhibitors (Figure 7E). The observations in this section confirm the ATR and ATM dependence of p38α/MK2 activation, but also uncover the existence of additional facets in checkpoint regulation that will require additional studies to delineate.

## 4. Discussion

The results summarized in the previous section uncover, for the first time, a role for the p38α/MK2 pathway in the regulation of the G_2_-checkpoint in cells exposed to IR.

It is significant that this role remains undetectable when examined by direct inhibition of p38α or MK2 kinase activity but comes to the fore after inhibiting Chk1. The contribution of the p38α/MK2 pathway to the G_2_-checkpoint increases markedly when specific Chk1 inhibitors are utilized rather than UCN-01, a result that can now be rationalized by the considerable effect of UCN-01 on MK2 [18]. The unique ability of the p38α/MK2 pathway to function only as a backup of Chk1 explains why its contribution remained undetected in previous studies analyzing activation of the G_2_-checkpoint in cells exposed to IR [18,20].

The p38α/MK2 pathway was first implicated in the regulation of the G_2_-checkpoint in studies on cellular responses to UV radiation [20]. The authors reported that p38α and p38β activities are required for the initiation of the G_2_-checkpoint, acting by phosphorylating Cdc25B, thus inducing binding to 14-3-3 proteins and its exclusion from the nucleus [20]. This UV-induced checkpoint was found to be resistant to caffeine and UCN-01 suggesting that ATM, ATR, and Chk1 were not involved (see also below).

Subsequent work convincingly demonstrated that MK2, rather than p38α, directly phosphorylates Cdc25A, Cdc25B, and Cdc25C in response to UV exposure and induces 14-3-3 protein binding and cytoplasmic translocation, or proteasomal degradation, to activate the S-phase and the G_2_-checkpoint responses [17]. However, in all cases, p38α activity was required to activate MK2, which could thus be hierarchically placed at the same checkpoint signaling position as Chk1 and Chk2 [17].

Importantly, p38 and MK2 form a tight complex within cells, and indeed, genetic work in mice shows that although p38 activates many kinases, MK2 is the key kinase responsible for p38-dependent biological processes [60]. In response to cellular stress, p38 phosphorylation of MK2 is required for the transport of the p38/MK2 complex from the nucleus to the cytoplasm [61,62]. Thus, MK2 functions as the nuclear initiator of Cdc25B and Cdc25C phosphorylation in response to DNA damage and the maintenance kinase that keeps Cdc25B and Cdc25C inhibited in the cytoplasm.

Further work demonstrated that p38α is activated in MMR-proficient, but not deficient, human glioma cells exposed to temozolomide (TMZ) [21]. Other works using UV-radiation demonstrate that in the absence of p53, the response of cells exposed to chemotherapeutic agents depends, in addition to Chk1/Chk2, also on the p38/MK2 pathway for cell-cycle arrest and survival after DNA damage [18]. Under these conditions, MK2 depletion in p53-deficient cells, but not in p53 wild-type cells, causes abrogation of the Cdc25A-mediated S-phase checkpoint after cisplatin exposure and loss of the Cdc25B-mediated G_2_-checkpoint following doxorubicin treatment [18]. Importantly, these responses are rather long-term responses, often detected many hours after exposure to DNA-damaging agents.

Notably, our work fails to detect dependence on p53 status in the Chk1-backup function of the p38α/MK2 pathway after exposure of cells to IR, suggesting that other or additional mechanisms are involved. In the above responses to chemotherapeutic agents [18], nuclear Chk1 activity is essential to establish a G_2_-checkpoint, while cytoplasmic MK2 activity is critical for prolonged checkpoint maintenance—through posttranscriptional mRNA stabilization upon translocation of the p38/MK2 complex from the nucleus to the cytoplasm, where it stabilizes Gadd45a mRNA by different mechanisms [19]. Our results, on the other hand, demonstrate that upon inhibition of Chk1, the p38α/MK2 pathway can also function in the early stages of checkpoint activation.

It is notable that one study reports that inhibition of MK2, in addition to Chk1 suppression, partially reverses the checkpoint abrogation [63], which contrasts our observations. While this may suggest that genetic context is important for the interplay among different checkpoint kinases, it is also relevant to keep in mind that our results reflect the response of G_2_-phase cells exposed to IR in the G_2_-phase of the cell cycle. This level of specificity in checkpoint analysis is unique in the present study and is guided by our previous work that detected diametrical changes in the crosstalk between ATM and ATR in G_2_-checkpoint regulation depending upon whether cells are irradiated during S- or G_2_-phase [13,14]. This may be another reason for the divergent results in the above study [63].

Additional work shows [22] that the p38α/MK2 and Chk1 pathways may have overlapping functions on G_2_-checkpoint activation and maintenance. Indeed, inhibition of both Chk1 and p38 pathways fails to cause greater bypass of TMZ-induced G_2_-arrest or greater cytotoxicity than inhibition of either pathway alone. This suggests that p38 and Chk1 are unlikely to bring about G_2_-arrest by reciprocal activation or phosphorylation. However, the Chk1 pathway was required for both the initiation and maintenance of TMZ-induced G_2_-arrest, whereas the p38 pathway performed a role only in its initiation.

Our results suggest that the activation of the p38α/MK2 pathway is dependent on ATM and ATR activity [18,19]. This is similar to responses reported for p38α/MK2 pathway activation in response to chemotherapeutic agents [18,19]. However, studies with Xenopus egg extracts and mammalian cells have pointed to the existence of a checkpoint pathway that is independent of ATR/Chk1 and ATM/Chk2 [20,21,22,23,24,64], and this is also true for cells exposed to UV [20,38]. Our results in Figure 7 also suggest ATM/ATR independent functions for the p38α/MK2 pathway. The ATM/ATR-independent block in cell cycle progression suggests further layers of checkpoint control that will require further investigations.

It is relevant to note that the results obtained in the present study with S-phase irradiated cells suggest that the role of the p38α/MK2 pathway in the regulation of the G_2_-checkpoint is similar to that of G_2_-phase irradiated cells. Thus, the shifts previously observed [13,14] seem to mainly affect crosstalk among the PI3-kinases sensing the DSB—possibly to regulate the space of DSB repair pathway choice, which in itself changes as cell progress from S-phase to G_2_-phase. Whether changes at this level of checkpoint signal initiation affect aspects of the regulation of the p38α/MK2 pathway remains to be elucidated.

Collectively, our results extend the functional spectrum of activities of the p38α/MK2 pathway by demonstrating that when Chk1 is inhibited, the backup function of the pathway extends to the regulation of checkpoint activation and that this function is p53 independent. The role of translocation from the nucleus to the cytoplasm of the p38α/MK2 complex in its Chk1 backup function requires further investigation. The results contribute to the initial impetus of the study, i.e., to determine why inhibition of ATR by VE-821 and of Chk1 by UCN-01 fail to give equivalent results. It is notable that the use of more specific Chk1 inhibitors actually shows that the backup contribution of the p38α/MK2 pathway is much stronger than originally thought. Retrospectively, it is now clear that the UCN-01 results published earlier reflect the inhibition of both Chk1 and MK2 [18].

## Figures and Tables

**Figure 1 cells-12-01387-f001:**
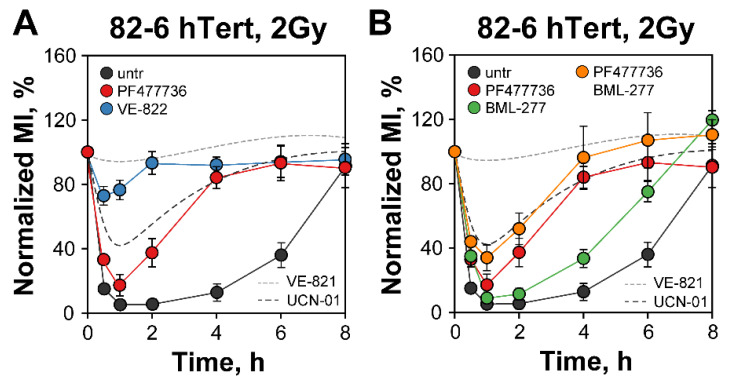
*Chk1, but not Chk2, inhibition partially abrogates the G_2_-checkpoint in irradiated cells.* (**A**) Effect of novel ATR inhibitor, VE-822 (ATRi), and Chk1 inhibitor, PF477736 (Chk1i), on G_2_-checkpoint in 82-6 hTert cells exposed to 2 Gy of IR. The normalized MI calculated as described in Materials and Methods is plotted as a function of time. The RAW MIs of non-irradiated cells at specific times after irradiation are shown in Appendix A. (**B**) Effects of single or combined inhibition of Chk1 (Chk1i) and Chk2, (BML-277), on 82-6 hTert cells exposed to 2 Gy. The effects of a previously utilized ATR inhibitor, VE-821, and the Chk1 inhibitor, UCN-01, are depicted with dashed lines. Untreated cells (untr) only receive DMSO. Data represent the mean and standard deviation (±SD) from three independent experiments.

**Figure 2 cells-12-01387-f002:**
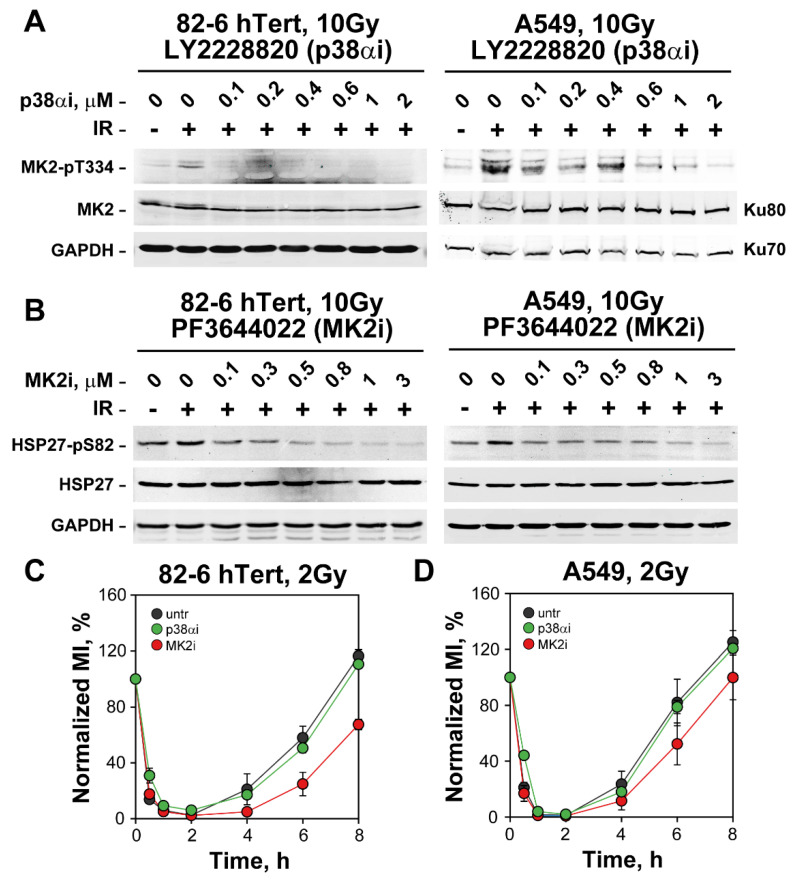
*Inhibition of p38α or MK2 leaves the G_2_-checkpoint intact in irradiated 82-6 hTert and A549 cells*. (**A**) Western blot analysis showing the effect of increasing concentrations of LY2228820, an inhibitor of p38α (p38αi), on 82-6 hTert and A549 cells exposed to 10 Gy and analyzed 1 h later. p38α activity is measured using the phosphorylation of its target, MK2, on threonine 334 (MK2-pT334) as a proxy. Total MK2 levels are also shown. The protein level of GAPDH, Ku70 and Ku80 serve as loading controls. (**B**) Western blot analysis showing the effect of PF3644022, an inhibitor of MK2 (MK2i), on cells exposed to 10 Gy and analyzed 1 h later. MK2 activity is measured using the phosphorylation of its target, HSP27 at Serine 82 (HSP27-pS82) as a proxy. Total HSP27 levels are also shown. GAPDH serves as a loading control. (**C**) As in Figure 1A after incubation with 1 µM p38αi or 1 µM MK2i. The ranges of MI values used for normalization are as follows: MI_untr_ = (1.63–2.15%), MI_p38αi_ = (1.81–2.6%) and MI_MK2i_ = (1.37–1.83%). (**D**) Same as in (**C**), but for A549 cells. The ranges of MI values used for normalization are: MI_untr_ = (1.68–2.49%), MI_MK2i_ = (1.84–2.39%) and MI_p38αi_ = (1.66–2.08%). The results in (**C**,**D**) represent the mean and standard deviation (±SD) from three independent experiments.

**Figure 3 cells-12-01387-f003:**
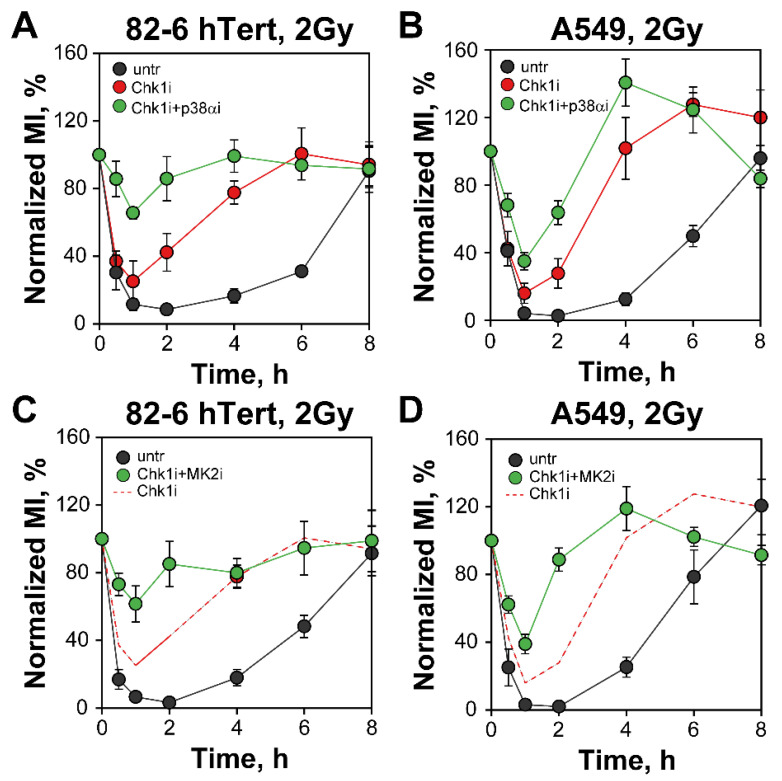
*Combined inhibition of Chk1 and p38α or MK2 abrogates the G_2_-checkpoint of 82-6 hTert and A549 cells exposed to 2 Gy or IR.* (**A**) As in Figure 1A for 82-6 hTert cells either left untreated or incubated with Chk1i, alone or in combination with p38αi. The ranges of MI used for normalization are: M_untr_ = (1.64–2.28%), MI_Chk1i_ = (1.85–2.56%), and MI_Chk1i + p38αi_ = (2.32–3.46%). (**B**) Same as (**A**), but for A549 cells. The ranges of MI used for normalization are: MI_untr_ = (1.69–3.23%), MI_Chki_ = (2.01–2.49%), and MI_Chk1i + p38αi_ = (2.02–2.88%). (**C**) As in (**A**), but for 82-6 hTert cells treated with combined Chk1i and MK2i. The ranges of MI used for normalization are: M_untr_ = (1.61–2.15%), and MI_Chk1i + MK2i_ = (2.1–2.9%). The red dashed line represents the Chk1i results in (**A**). (**D**) Same as (**B**), but for A549 cells treated with combined Chk1i and MK2i. The ranges of MI used for normalization are: MI_untr_ = (1.72–2.91%) and MI_Chk1i + MK2i_ = (1.87–3.07%). The red dashed line represents Chk1i results, shown in (**B**). All inhibitors were administrated to the cells 1 h prior to irradiation, at the following concentrations: Chk1i, 300 nM, p38αi,1 µM, and MK2i, 1 µM. Data represent mean and standard deviation (±SD) from three independent experiments.

**Figure 4 cells-12-01387-f004:**
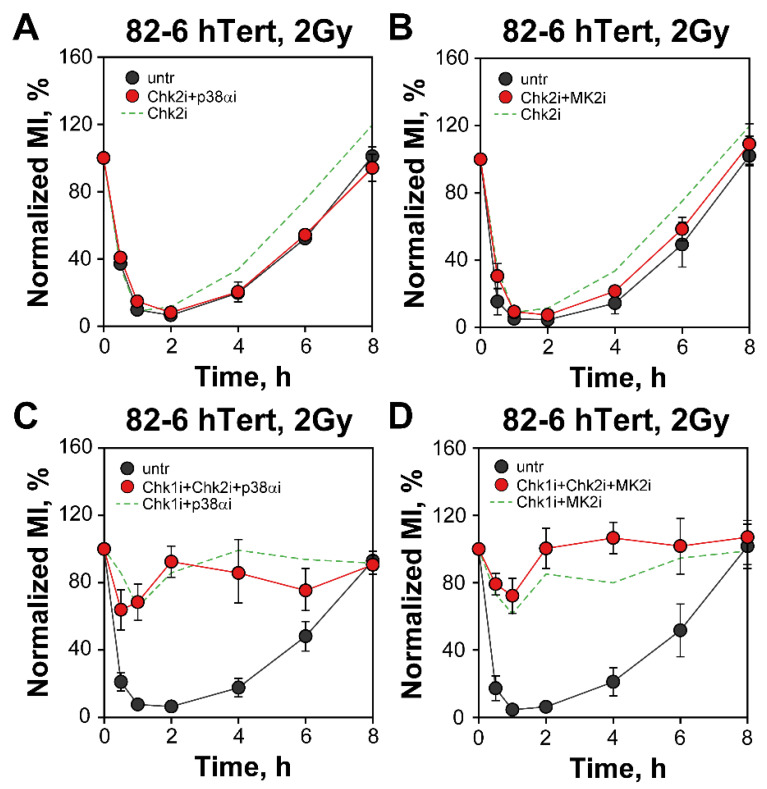
*Chk2i fails to enhance the G_2_-checkpoint effects of Chk1i, p38αi or MK2i.* (**A**) 82-6 hTert cells treated with Chk2i alone or combined with Chk2i and/or p38αi. The MI ranges used for normalization are: MI_untr_ = (1.65–2.17%) and MIChk2i + MK2i = (1.63–1.98%). The green dashed line represents Chk2i data plotted in Figure 1B. (**B**) 82-6 hTert cells treated with a combination of Chk2i and MK2i. The MI ranges used for normalization are: MI_untr_ = (1.63–2.38%) and MI_Chk2i + p38αi_ = (1.97–2.28%). The green dashed line represents Chk2i data shown in Figure 1B. (**C**) 82-6 hTert cells treated with a combination of Chk1i, Chk2i and p38αi. The MI ranges used for normalization are: MI_untr_ = (1.58–2.18%) and MI_Chk1i + Chk2i + p38αi_ = (1.98–2.94%). The green dashed line represents the results shown in Figure 3A. (**D**) 82-hTert cells treated with combinations of Chk1i, Chk2i and MK2i. The MI ranges used for normalization are: MI_untr_ = (1.61–2.15%) and MI_Chk1i + Chk2i + MK2i_ = (1.83–2.91%). The green dashed line represents the results of combined Chk1i and p38αi treatment shown in Figure 3C. Data represent mean and standard deviation (±SD) from three independent experiments.

**Figure 5 cells-12-01387-f005:**
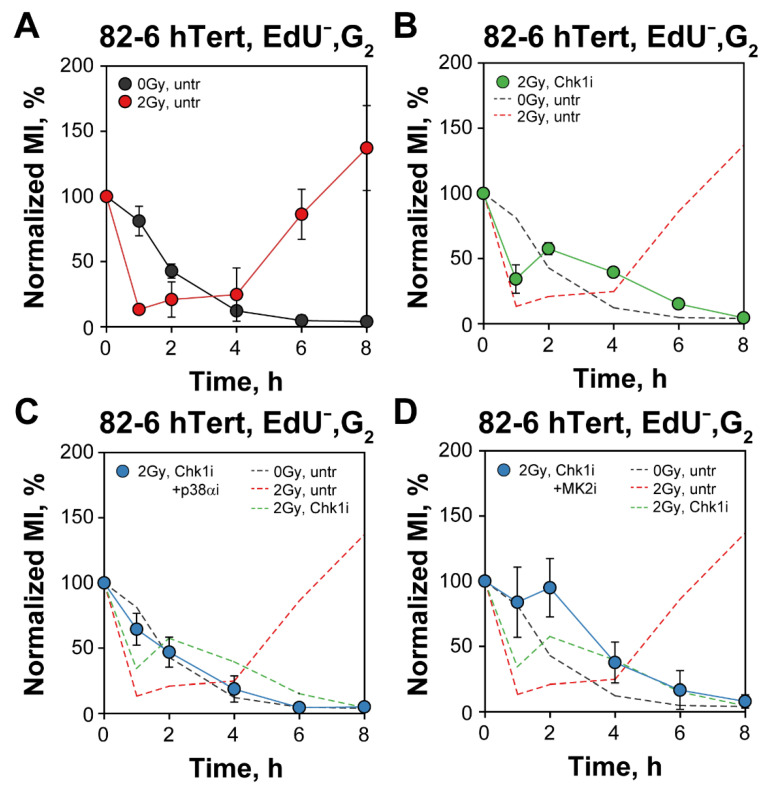
Three parametric flow cytometry analysis of G_2_-checkpoint confirms a p38α and MK2 dependent checkpoint activation when analyzing exclusively G_2_-phase irradiated cells after Chk1 inhibition. (**A**) Three parametric flow cytometry analysis showing the normalized MI as a function of time in EdU^−^, G_2_-cells, in non-irradiated or 2 Gy irradiated 82-6 hTert cells. (**B**) As in (**A**) for samples treated with Chk1i. The dashed lines depict the results shown in (**A**) for comparison. (**C**) As in (**B**) for samples treated with both Chk1i and p38αi. The dashed lines are transferred from (**A**,**B**) and are shown for comparison. (**D**) As in (**C**), but for cells irradiated in the presence of both Chk1i and MK2i. See text for details on methodology and interpretation. Data represent mean and standard deviation (±SD) from three independent experiments.

**Figure 6 cells-12-01387-f006:**
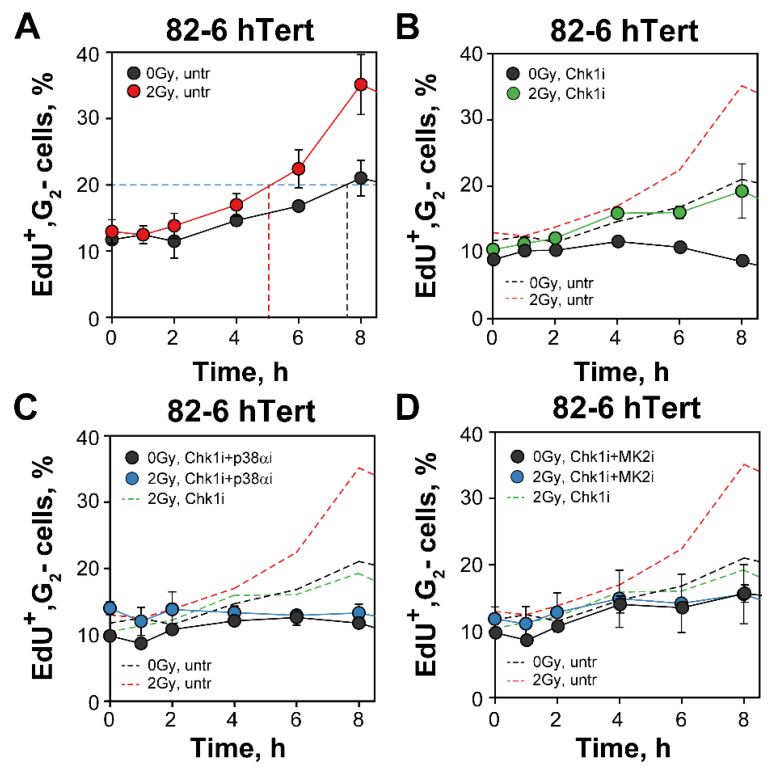
*Three parametric flow cytometry analysis of G_2_-checkpoint reveals a p38α/MK2 dependent activation also in S-phase irradiated cells following Chk1 inhibition.* (**A**) Percentage of EdU^+^ cells, representing the fraction of S-phase cells at the time of irradiation with 2 Gy, entering the G_2_-phase as a function of time thereafter. Results with unirradiated cells are also shown. The dashed line is arbitrarily drawn at 20% to help estimate the radiation-induced delay. (**B**) As in (**A**) for cells treated with Chk1i. The dashed lines depict the results in (**A**) and are shown for comparison. (**C**) As in (**B**) for cells treated with both Chk1i and p38αi. (**D**) As in (**C**) for cells treated with Chk1i and MK2i. See text for details on methodology and interpretation. Data represent mean and standard deviation (±SD) from three independent experiments.

**Figure 7 cells-12-01387-f007:**
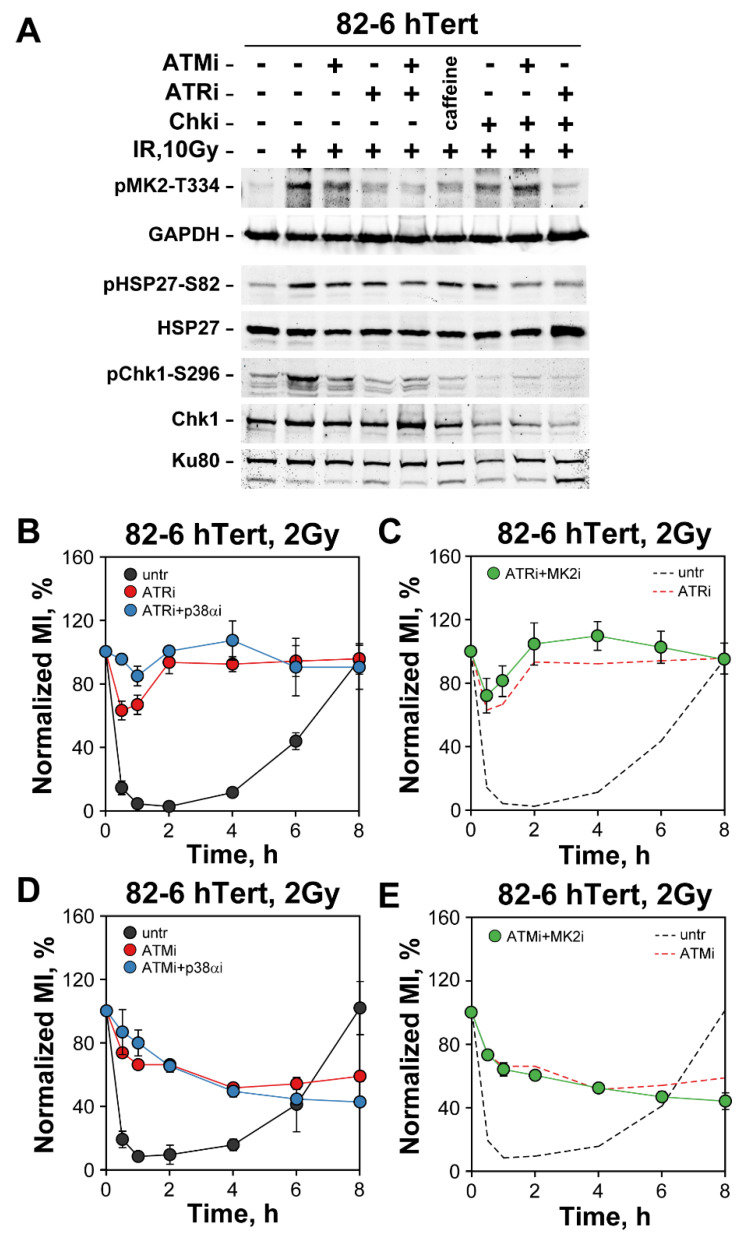
*Roles of ATM and ATR in the activation of p38α/MK2 pathway and the G_2_-checkpoint.* (**A**) Western blot analysis of pMK2-T334 and pHSP27-S82 after treatment of 82-6 hTert cells with the indicated PIKK inhibitors. GAPDH, HSP27, Ku80 and Chk1 served as loading controls. (**B**) G_2_-checkpoint analysis in 82-6 hTert cells treated with ATRi, or a combination of ATRi and p38αi. The range of MI are: MI_untr_ = (1.61–2.14%), MI_ATRi_ = (2–2.92%), and MI_ATRi + p38αi_ = (2.02–2.88%). (**C**) G_2_-checkpoint analysis in 82-6 hTert cells treated with a combination of ATRi and MK2i. The ranges of MI are MI_ATRi + MK2i_ = (1.6–2.87%). (**D**) G_2_-checkpoint analysis in 82-6 hTert cells treated with Ku55933 (ATMi) or a combination of ATMi and p38αi. The ranges of MI are: MI_untr_ = (1.61–2.14%), MI_ATMi_ = (1.68–1.84%), and MI_ATMi + p38αi_ = (1.76–2.93%). (**E**) G_2_-checkpoint analysis in 82-6 hTert cells treated with a combination of ATMi and MK2i. The range of MI is MI_ATMi + MK2i_ = (1.78–2.67%). Dashed lines in (**C**,**E**) illustrate data presented in (**B**,**D**), respectively, and are shown for comparison. All data represent means and standard deviation (±SD) from two or three independent experiments.

## Data Availability

The flow cytometry datasets generated by two and three parametric flow cytometry analysis are available upon request from the corresponding author.

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
