# Peer review of "The p38/MK2 Pathway Functions as Chk1-Backup Downstream of ATM/ATR in G2-Checkpoint Activation in Cells Exposed to Ionizing Radiation"

_cells, 2023, doi:10.3390/cells12101387_

Round 1

Reviewer 1 Report

The manuscript by D. Luo et al. addresses mechanisms of the IR-induced G2 checkpoint. By the use of kinase inhibitors of ATR, CHK1, CHK2, MK2 and p38alpha, it is shown that the IR-induced G2 checkpoint is almost completely dependent on ATR but only partially dependent on CHK1. However, when CHK1 is inhibited, the IR-induced G2 checkpoint becomes dependent on p38alpha/MK2. The p38alpha/MK2 pathway thus appears to act as a backup to CHK1 signaling after IR. The results are interesting, but a few issues are unclear and need to be improved.

Major issues

1.     When making conclusions based on kinase inhibitors with different targets, the choice of inhibitor concentration is critical. Does each inhibitor inhibit its respective target to the same extent? The manuscript contains some results showing that ATRi inhibits CHK1-p-S345 and CHK1i inhibits CHK1-pS296 (Figure S1CD). However, in S1C & S1D different cell lines are used for ATRi or CHK1i. This is a problem as it is not clear whether the extent of inhibition will be similar in both cell lines. Furthermore, densitometry analysis of the blot shown in Figure S1D is not sufficient to demonstrate that 300nM of CHK1i inhibits CHK1-p296 by 90%. What is the densitometry reading if 10% of the sample in lane 2 (“IR+, CHK1i 0”) is loaded?

2. To better quantify the extent of target inhibition (Fig S1C, S1D, Fig 2AB), a dilution series should be included of the sample treated with IR alone.  When a dilution series is included in the blot, f.ex. by loading 100%, 50%, 25%, 10% and 5% of the sample treated with IR alone, the densitometry readings of the samples treated with IR+inhibitor (lanes 3-8) can be compared to the densitometry readings of the dilution series.

3.     Figures 1, 2 & 3. It is not clear how MI% was normalized. What are the effects of the inhibitors alone (without IR)? Could these effects also be shown, at least in a supplementary figure? The MI values described for normalization in the figure legends are unclear. How does the MI vary for untreated cells at 0-8 hours? And how does the MI vary at 0-8 hours after each inhibitor treatment?

4.      The conclusion that the IR-induced G2 checkpoint becomes dependent on p38alpha/MK2 when CHK1 is inhibited, is interesting. -Is MK-pT334 increased after CHK1i?

5.     Does ATR phosphorylate both CHK1 and p38alpha/MK? Is MK-pT334 reduced after ATRi?

Minor issues:

6.     Lines 518-526. It is unclear from the text whether Ref 63 used IR. A discussion of IR versus other DNA damaging might be included here.

7.     Abstract lines 16-22: The introductory part of the abstract is long and complicated. The strong focus on DNAPk and ATM in the beginning of the abstract seems a bit misleading with respect to the content of this manuscript.

Author Response

Response to the Reviewer’s comments

 Cells

“The p38/MK2 pathway functions as Chk1-backup downstream of ATM/ATR in G2-checkpoint activation in cells exposed to ionizing radiation”

Luo et al.

cells-2327335

Response to the Review Report 1

“The manuscript by D. Luo et al. addresses mechanisms of the IR-induced G2 checkpoint. By the use of kinase inhibitors of ATR, CHK1, CHK2, MK2 and p38alpha, it is shown that the IR-induced G2 checkpoint is almost completely dependent on ATR but only partially dependent on CHK1. However, when CHK1 is inhibited, the IR-induced G2 checkpoint becomes dependent on p38alpha/MK2. The p38alpha/MK2 pathway thus appears to act as a backup to CHK1 signaling after IR. The results are interesting, but a few issues are unclear and need to be improved.”

We thank the reviewer for highlighting the essence of the current paper and for recognizing our contribution to the field by revealing a novel, p38a/MK2 dependent checkpoint mechanism sustaining the G2-checkpoint under conditions inhibiting the Chk1 effector kinase.

In the revised manuscript, we made every effort to address points and concerns raised by the reviewer and explain in detail the rationale of our response. We hope that by doing so, we improved significantly the scientific quality of our study and bring clarity to our working hypothesis.

Major comments:

“1) When making conclusions based on kinase inhibitors with different targets, the choice of inhibitor concentration is critical. Does each inhibitor inhibit its respective target to the same extent? The manuscript contains some results showing that ATRi inhibits CHK1-p-S345 and CHK1i inhibits CHK1-pS296 (Figure S1CD). However, in S1C & S1D different cell lines are used for ATRi or CHK1i. This is a problem as it is not clear whether the extent of inhibition will be similar in both cell lines. Furthermore, densitometry analysis of the blot shown in Figure S1D is not sufficient to demonstrate that 300nM of CHK1i inhibits CHK1-p296 by 90%. What is the densitometry reading if 10% of the sample in lane 2 (“IR+, CHK1i 0”) is loaded?

We thank the reviewer for highlighting the significance of inhibitor concentration in interpreting our results. We completely agree with the reviewer that the inhibitor concentration is crucial for the effect observed, especially when investigating different cell lines. To address this concern, we have conducted additional experiments using 82-6 hTert and A549 cells to evaluate the phosphorylation levels of specific Chk1 serine residues (Serine 345 and Serine 296), which are used as a surrogate for ATR and Chk1 activation, respectively. The results of these experiments are now included in Figure S1. In addition, in the revised manuscript, we refrain from quantitative outlines unless we have quantitative data.

 “2) To better quantify the extent of target inhibition (Fig S1C, S1D, Fig 2AB), a dilution series should be included of the sample treated with IR alone.  When a dilution series is included in the blot, f.ex. by loading 100%, 50%, 25%, 10% and 5% of the sample treated with IR alone, the densitometry readings of the samples treated with IR+inhibitor (lanes 3-8) can be compared to the densitometry readings of the dilution series.” 

We appreciate the reviewer's suggestion. However, in the present study, we employed a semi-quantitative western blot analysis to determine the working concentrations of the selected checkpoint inhibitors. Our goal was to identify the concentrations that provide strong inhibition of the targeted protein kinase, while minimizing disruption to the cell cycle. We note that the experimentally determined inhibitor concentrations fall within the range of concentrations extensively used in the literature in a plethora of cell lines, such as ATRi, VE-822 (80-400nM), Chk1i, PF477736 (45-300nM), p38αI, LY2228820 (0.2-1 μM), and MK2i, PF3644022 (0.16-10 μM). While some studies have used lower inhibitor concentrations but longer incubation times (e.g., 24 hours), we refrained from using such protocols, as they may cause shifts in cell cycle distribution. To minimize such effects, in our study, we administered the inhibitors 1 h prior to irradiation and kept them in the growth media until collecting the samples for analysis at the indicated times after irradiation.

“3) Figures 1, 2 & 3. It is not clear how MI% was normalized. What are the effects of the inhibitors alone (without IR)? Could these effects also be shown, at least in a supplementary figure? The MI values described for normalization in the figure legends are unclear. How does the MI vary for untreated cells at 0-8 hours? And how does the MI vary at 0-8 hours after each inhibitor treatment?”

In our experimental setup, the impact of the inhibitors on non-irradiated cells was negligible. Therefore, we did not include graphs of the non-irradiated cells in the corresponding figures. We provide, instead, the RAW MI in the figure legends, which allows the reader to see the effects of the inhibitors on the MI. Nevertheless, to increase the clarity of the manuscript, we have incorporated the requested information in the revised version of the manuscript. Figures S2A and S2B summarize the requested data on the RAW MI of non-irradiated cells.

“4) The conclusion that the IR-induced G2 checkpoint becomes dependent on p38alpha/MK2 when CHK1 is inhibited, is interesting. -Is MK-pT334 increased after CHK1i?”

Our newly conducted Western blot analysis does not reveal any increase in MK2 phosphorylation at T334 after inhibition of Chk1. Moreover, there is a slight decrease in the pMK2-T334 signal in 82-6 hTert cells and almost no change in A549 cells after inhibition of Chk1. These results are now incorporated in the new Figure 7A and Figure S5C.

“5) Does ATR phosphorylate both CHK1 and p38alpha/MK? Is MK-pT334 reduced after ATRi?”

In response to this point, we conducted experiments investigating the contribution of ATM and ATR to the phosphorylation of p38/MK2. Specifically, we examined the levels of pMK2-T334 and pHSP27-S82 after IR exposure of 82-6 hTert and A549 cells in the presence of ATM and ATR inhibitors. The newly obtained results are incorporated in Figure 7A and Figure S5 of the revised manuscript. They show that ATR is involved in the phosphorylation of p38 and MK2. However, the additional decrease of the pMK2-T334 observed after combined treatment with ATMi and ATRi suggests that ATM is also involved – presumably, the ATM/ATR module plays a role in p38 activation. This is further supported by the observation that phosphorylation of the MK2-specific target pHSP27-S82 is down regulated after both ATMi and ATRi treatment. These findings provide valuable insights into the signalling mechanisms controlling the G2-checkpoint and the role of p38 in this regulation.

Minor issues:

“6) Lines 518-526. It is unclear from the text whether Ref 63 used IR. A discussion of IR versus other DNA damaging might be included here.”

We have included a section briefly discussing the effects in cells exposed to other DNA-damaging agents, as requested by the Reviewer.

“7) Abstract lines 16-22: The introductory part of the abstract is long and complicated. The strong focus on DNAPk and ATM in the beginning of the abstract seems a bit misleading with respect to the content of this manuscript.”

In the revised version of the manuscript, we have shortened and simplified the abstract by removing references to DNA-PKcs.

Reviewer 2 Report

Review on the manuscript by D. Luo et al.

In their submission to Cells (MDPI), D. Luo and co-authors searched for backup pathways that trigger a G2 checkpoint response in irradiated cells in the presence of Chk1 inhibition.

Using more specific CHK1 inhibitors, they specified their earlier CHK1 UCN-01 inhibitor results and observed that CHK1 inhibition only moderately relaxes the G2 checkpoint. Their new results unveil that a strong residual G2-checkpoint following Chk1 inhibition is dependent on the p38α/MK2 pathway. Based on the current results, the authors propose that in IR-exposed cells with inhibited Chk1 signaling, the p38α/MK2 pathway is acting as a back-up pathway at the G2 checkpoint, a highly significant finding.

The paper is well written and the experiments carried out are state of art. However, the details of the experimental procedures used are obscure at some points to the non-specialist reader. The Methods section needs more details on the experimental procedures, especially a table showing all the antibody dilutions used, as recommendations of manufacturers are usually ranges of dilutions.

The results obtained show a novel role for the p38/MK2 pathway in the regulation of the G2-checkpoint in response to ionizing radiation. Thus, this contribution adds new data and fits well into the scope of ‘Cells’. A few minor suggestions for improvement are given below.

L59: “They interphase with” - should this mean "interact with" ?

L66: Pls specify ‘low doses’ in terms of Gy. The term “low dose” has a wide meaning…

L79: ...shepherded ?

L135 & below: Mention briefly how the concentrations used were determined, e.g. by referring to the corr. suppl. Figs.

2.3.: Indicate the IR doses applied. Please mention which dosimeter was used to measure the doses in the instrument.

 2.4. 2.5.: Please add a table showing or mention the dilutions used for all the antibodies in these sections. This is important, since recommendations of Ab manufacturers for primary Abs are usually ranges of dilutions. This will enhance reproducibility (see L160,182, 201).

 L172: write out the abbreviations PI, EdU at first occurrence.

L176: "After irradiation" – mention the time frame of the time courses performed after irradiation.

L176: mention the applied IR doses.

L185: Please specify whether the binding of PI to RNA played a role in your FCM measurements and whether your staining protocol involved RNAase treatment.
Mention the manufacturer of PI and the concentration of PI used in the staining reaction.

L218-221: Sentence too long and complex. Please shorten and simplify.

L224: Add reference to this statement.

Lines 206 – 234 recapitulate the introduction. Thus, it is suggested to shorten this section.

 L254: Write out "MI" at this point, to facilitate the access to the figure.

 L270: mention the exact doses used here.

L271: "of ATR signal" – of the ATR signal (?)

L284: Is this a statistically significant observation?

L317: western – put idiom in capitals: ‘Western’

L491: Not clear to which point in the argumentation line "Here" refers to. Please clarify.

497: ... work demonstrates

L505: mention that your work deals with ionizing radiation

L522: ... cells exposed to ...

L547: … further layers of ...

Author Response

Response to the Reviewer’s comments

 Cells

“The p38/MK2 pathway functions as Chk1-backup downstream of ATM/ATR in G2-checkpoint activation in cells exposed to ionizing radiation”

Luo et al.

cells-2327335

Response to the Review Report 2

“In their submission to Cells (MDPI), D. Luo and co-authors searched for backup pathways that trigger a G2 checkpoint response in irradiated cells in the presence of Chk1 inhibition.

Using more specific CHK1 inhibitors, they specified their earlier CHK1 UCN-01 inhibitor results and observed that CHK1 inhibition only moderately relaxes the G2 checkpoint. Their neu results unveil that a strong residual G2-checkpoint following Chk1 inhibition is dependent on the p38α/MK2 pathway. Based on the current results, the authors propose that in IR-exposed cells with inhibited Chk1 signaling, the p38α/MK2 pathway is acting as a back-up pathway at the G2 checkpoint, a highly significant finding.

The paper is well written and the experiments carried out are state of art. However, the details of the experimental procedures used are obscure at some points to the non-specialist reader. The Methods section needs more details on the experimental procedures, especially a table showing all the antibody dilutions used, as recommendations of manufacturers are usually ranges of dilutions.

The results obtained show a novel role for the p38/MK2 pathway in the regulation of the G2-checkpoint in response to ionizing radiation. Thus, this contribution adds new data and fits well into the scope of ‘Cells’. A few minor suggestions for improvement are given below.”

We appreciate the reviewer's recognition of the significance of the topic of our paper and the constructive comments.

“1) L59: “They interphase with” - should this mean "interact with" ?”

Passage edited.      

“2) L66: Pls specify ‘low doses’ in terms of Gy. The term “low dose” has a wide meaning…”

The dose is defined as suggested

“3) L79: ...shepherded ?”

This expression is corrected.

“4) L135 & below: Mention briefly how the concentrations used were determined, e.g. by referring to the corr. suppl. Figs.”

A reference to the utilized inhibitors concentration, with the explanation of the rationale behind the utilized concentration values, is now included at the indicated paragraph in the revised manuscript.

“5) Indicate the IR doses applied. Please mention which dosimeter was used to measure the doses in the instrument.”

The requested information is now provided.

“6) Please add a table showing or mention the dilutions used for all the antibodies in these sections. This is important, since recommendations of Ab manufacturers for primary Abs are usually ranges of dilutions. This will enhance reproducibility (see L160,182, 201).

A supplementary table, incorporating the requested information, is compiled and included as a Supplementary Table 1.

“7) L172: write out the abbreviations PI, EdU at first occurrence.

Abbreviations are defined at their first appearance in the revised version of the manuscript.

“8) L176: "After irradiation" – mention the time frame of the time courses performed after irradiation.”

The time points at which cells are collected after irradiation are indicated in the revised Materials and Methods sections.

“9) L176: mention the applied IR doses.”

The IR doses used are now given in the edited version of the manuscript.

“10) L185: Please specify whether the binding of PI to RNA played a role in your FCM measurements and whether your staining protocol involved RNAase treatment. Mention the manufacturer of PI and the concentration of PI used in the staining reaction.

These issues are now clarified.

“11) L218-221: Sentence too long and complex. Please shorten and simplify.”

Passage edited.

“12) L224: Add reference to this statement.”

The requested reference is added.

“13) Lines 206 – 234 recapitulate the introduction. Thus, it is suggested to shorten this section.”

The indicated section is shortened as suggested.

“14) L254: Write out "MI" at this point, to facilitate the access to the figure”

We followed the suggestion of the Reviewer and provide this information throughout.

“15) L270: mention the exact doses used here.”

Doses are now specified.

“16) L271: "of ATR signal" – of the ATR signal (?).”

Passage corrected.

“17) L284: Is this a statistically significant observation?”

The difference in the effect of Chk1 on the MI inhibition between 82-6 hTert and A549 cells is small. However, the difference in the response at the latter times after IR (2, 4, 8h) is more prominent than the effect observed at 0,5 and 1h. We have rephrased the corresponding sentence to reflect better the presented results.

“18) L317: western – put idiom in capitals: ‘Western’.”

Corrected according to the reviewer's suggestion

“19) L491: Not clear to which point in the argumentation line "Here" refers to. Please clarify.”

Passage clarified.

“20) L497: ... work demonstrates.”

Corrected.

“21) L505: mention that your work deals with ionizing radiation.”

The focus of our paper is mentioned at this location as suggested.

“22) L522: ... cells exposed to ...”

Corrected.

“23) L547: … further layers of ...”

Corrected.

Round 2

Reviewer 1 Report

All my concerns have been addressed.